https://doi.org/10.5194/egusphere-2025-2708



# Historical Droughts in British Colonial Belize (1771-1981)

Oriol Ambrogio Gali[1], Sarah E. Metcalfe[1], Elizabeth A. C. Rushton[2], Betsabé de la Barreda-Bautista[1], Georgina H. Endfield[3], Sofia Márdero[1], Franziska Schrodt[1], Alec McLellan[4]

[1]School of Geography, University of Nottingham, University Park, NG7 2RD, UK
[2]Education Division, University of Stirling, Stirling, FK9 4LA, UK
[3]Department of History, University of Liverpool, 12 Abercrombie Square, L69 7WZ, UK
[4]Department of Anthropology, University of Cincinnati, 481 Braunstein Hall, OH 45221-0380, USA

*Correspondence to*: Sarah E. Metcalfe (sarah.metcalfe@nottingham.ac.uk)

**Abstract.** Belize, located on the Caribbean coast of the Yucatan peninsula, is increasingly vulnerable to hydroclimatic hazards such as droughts, which have caused widespread agricultural losses, water shortages, and economic disruption in recent years. Despite these risks, long-term climate reconstructions for the country remain lacking. This study presents the first documentary-based reconstruction of droughts in British colonial Belize from 1771 to 1981, using a diverse body of unpublished and published sources including newspapers, missionary letters, agricultural reports, and early
instrumental records. Droughts were identified through both direct meteorological references and indirect evidence such as crop failures, forest fires, and water scarcity, and were classified by severity and confidence levels. Results show that droughts were more frequent, longer, and more severe in the northern districts. The wetter southern districts experienced fewer and less intense droughts. Instrumental data partially corroborate the documentary findings, but also reveal key discrepancies, particularly for the pre-20th-century period. Comparison with drought records from the Mexican Yucatán
Peninsula, Guatemala, and the Caribbean suggests some regionally synchronous events, alongside droughts that appear specific to Belize. By extending the climate record back two centuries, this study provides critical historical context for current and future drought trends in Belize and the wider region. It highlights the importance of combining documentary and instrumental sources to assess long-term climate variability in data-scarce tropical environments and contributes to broader efforts to understand past climate extremes in the context of growing climate risk.

## 1 Introduction

Belize is located on the Caribbean coast of the Yucatan peninsula (Fig. 1a). Present-day Belize was partially occupied by the Spanish empire during the mid-16th century and later attracted British loggers in the 1710s, who came to exploit its timber resources. Initially, settlements were concentrated around present-day Belize City and in the north. The area
became known as British Honduras and was raised to the status of a Crown Colony with its own Governor in 1871 (Jones, 1989; Restall, 2019). Unlike other British colonies in the Caribbean, the economy of British Honduras was primarily based on forestry, with agriculture remaining a secondary activity until the early 20th century (Dobson, 1973; Bolland, 1977; Chalmin, 1990). The country was renamed Belize in 1973 and gained independence in 1981.

Belize's tropical climate, with distinct dry and wet seasons, is primarily influenced by the Intertropical Convergence Zone
(ITCZ) and the Caribbean Low-Level Jet (CLLJ). The ITCZ, especially significant in southern Belize, moves northward during the northern hemisphere summer, bringing heavy rainfall. The wet season generally extends from June to November, while the dry season spans from December to May. The transition from the wet to dry season is gradual. From November to February, frontal systems known as 'nortes' can bring occasional rainfall, which can be significant for local water availability and agriculture. Tropical storms and hurricanes, which move westward through the Caribbean, are most
frequent from June to November, with peak activity in September and October. Rainfall varies significantly between



Belize's northern and southern districts (Fig. 1b). In the northern town of Corozal, annual precipitation ranges between 52.7 and 60.9 inches [1339 to 1547 mm], whereas in the southern settlement of Punta Gorda, it ranges from 120.9 to 151.5 inches [3072 to 3848 mm]. The dry season in southern Belize is typically shorter, lasting from February to April. The wet season begins as early as May in the southernmost district of Toledo, while in the northernmost district of Corozal,

it typically starts in early June. A notable reduction in rainfall often occurs in August during the wet season, a phenomenon known as the 'Mauger season', although this pattern is less pronounced in the Toledo District. The 'Mauger season' is comparable to the 'canícula', a midsummer drought that affects Mexico from mid-July to late August (Walker, 1973; National Meteorological Service Belize; Magaña et al., 1999; Vanzie, 2008; Climate Studies Group Mona, 2020). These drier conditions in the midsummer months are driven by the CLLJ, which causes increasing moisture divergence, reducing

rainfall over parts of the Caribbean.

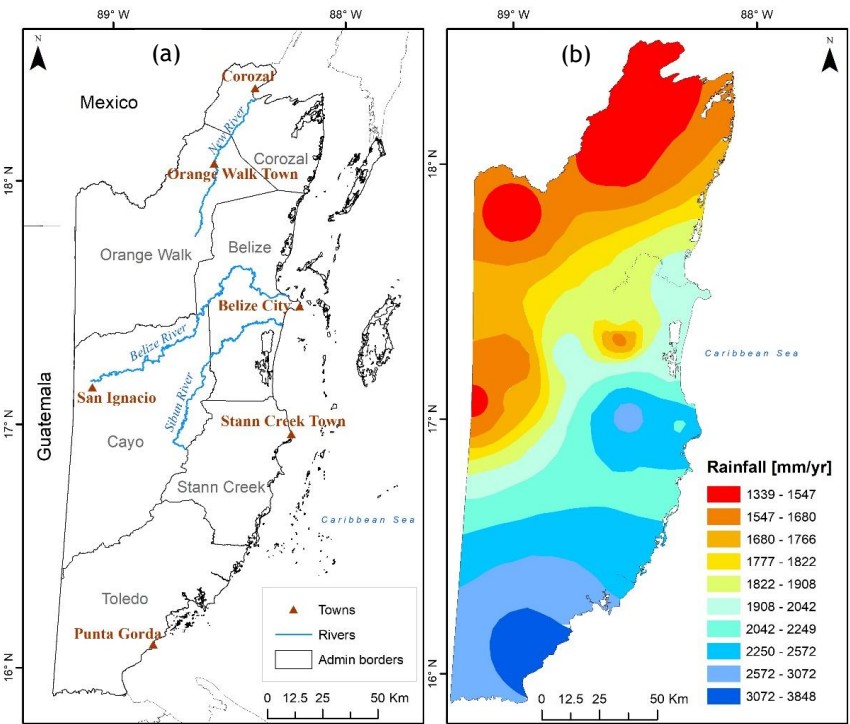

**Figure 1 (a): Map of Belize showing major towns, rivers and the administrative boundaries of its six districts. (b): Mean annual rainfall (mm) over Belize based on data from 1951 to 2013** (adapted from Frank Tench, https://belize.com/belize-annual-rainfall/).

The country is highly vulnerable to climate related hazards such as droughts, hurricanes, floods and sea level rise (International Monetary Fund, 2018). Droughts, in particular, pose a recurrent challenge especially in the northern districts, where precipitation levels are lower compared to other parts of the country (Climate Resilient Food Systems Alliance, 2022). Prolonged dry spells can have devastating effects on the local economy, agricultural development and employment levels. For instance, the severe drought of 2019, during which the International Airport meteorological

station recorded the lowest rainfall since 1952 (49.03 inches [1,245.6 mm]), resulted in a significant economic slowdown, severely impacting agricultural output and hydropower generation. The dry conditions halved maize production and citrus output fell because of citrus greening diseases. More than 7 thousand farmers were affected by the drought, mainly in the



northern half of the country and at least 418,000 acres [169,158.6 ha] of crops were lost. The drought also generated a rise in food price inflation and contributed to the rise in the unemployment rate by almost 3 % (Caribbean Development Bank, 2019; International Federation of Red Cross and Red Crescent Societies, 2019).

Investigating past climates is crucial for contextualising future climatic shifts within a long-term temporal framework (Mock, 2007). While some research has examined rainfall variability and historical droughts in the Caribbean (Mendoza et al., 2007; Berland et al., 2013), no such studies have focused on Belize, leaving a significant gap in our understanding of this region's climate history. The aim of this article is to extend the climate record beyond the limited instrumental data available and to better assess the long-term patterns of droughts, especially before the start of reliable instrumental observations. This is achieved by creating the first documentary-derived chronology of droughts of British colonial Belize from 1771 to 1981 using both published and unpublished sources. Early instrumental observations from various meteorological stations are also used to verify and complement the documentary reconstruction, providing better geographical coverage of the chronology. The similarities and discrepancies between the documentary and instrumental sources are then analysed, followed by a comparison of historical droughts in British colonial Belize and neighbouring regions. Finally, drought intensity and duration during the study period is compared with estimates for the Terminal Classic period (800 – 1000 CE) and future climate projections, to provide a longer term perspective on drought and its impacts.

## 2 Instrumental meteorological records

Instrumental data, primarily comprising precipitation and temperature records, are available starting from the second half of the 19th century. The earliest instrumental observations date back to 1848 and provide monthly precipitation and temperature data for present-day Belize City. However, the exact location of the meteorological station and the methods used are unknown (Temple, 1857). The first official meteorological station was established at the Public Hospital in Belize in 1863, with records maintained by the Colonial Surgeon (Rushton, 2014). Although rainfall records for Belize City have continued to the present day, the observatories have changed locations and probably methodologies several times. By the end of the 19th century, meteorological stations were also established in the Corozal, Toledo, and Stann Creek districts. Several stations were set up in the second half of the 19th century on private estates, and their records were usually published in local newspapers. These records, however, cover only brief periods, either because the measurements were soon discontinued or the records ceased to be published.

By the early 20th century, every district had at least one official meteorological station and their numbers further increased between 1933 and 1949 with the establishment of observatories in the Agricultural stations in the districts of Corozal, Orange Walk, Cayo, Stann Creek and Toledo. The number of weather stations at the end of the 1950s was reported to be 44, while in 1970 only 35 were active and recorded precipitation (Wright et al., 1959; Walker, 1973). In the 1970s, the amount of rainfall data available reduced considerably. By 1972, precipitation data were available from only 14 stations across the country. This number dwindled to 8 by 1976 and further decreased to 7 by 1981, the year of independence and the conclusion of the study period. Notably, weather stations in the northern districts of Corozal and Orange Walk vanished from records in the 1970s, only reappearing in 1981 (British Honduras, Department of Agriculture, 1973; British Honduras, Department of Agriculture, 1977; Ministry of Natural Resources, Department of Agriculture, 1982). Since 1981, the number of weather stations has continued to be very variable and data collection often quite problematic.



Caution is essential when interpreting the precipitation series from British Honduras, particularly the earliest records, which often contain discrepancies that underscore their problematic nature. A striking example is the comparison between annual precipitation totals recorded by the Public Hospital and St. Joseph's Observatory in Belize City during the 1880s and early 1890s. Despite being located just 400 yards [365 m] apart, the data reveal significant differences, probably resulting from flawed methodologies or faulty instrumentation at one of the stations. For instance, in 1888, the Public Hospital reported an annual rainfall of 77.08 inches [1957.8 mm], whereas St. Joseph's Observatory recorded a total of 128.15 inches [3255 mm]. Similar disparities were observed in 1891, with the Public Hospital reporting 56.82 inches [1443.2 mm] and St. Joseph's Observatory reporting 71.19 inches [1808.2 mm] (The Angelus, 1887-1900; Walker, 1973). Colonial sources frequently showed their scepticism about the reliability of instrumental data. For instance, the 1890 Annual Report raised doubts about the accuracy of the reported mean temperature of 56 °F [13.3 °C] for March in Belize City, deeming it "of doubtful credibility" (Great Britain, Parliament, House of Commons, 1891). Likewise, during the reorganisation of the colony's precipitation recording system in 1930, numerous rain gauges were found to be "in a deplorable condition", suggesting that "the records may not be entirely reliable" (Pim, 1934).

The annual rainfall totals from 13 stations with the longest precipitation records across Belize's six districts (ten stations in the north and four in the south), have been compiled. These stations were chosen for their long precipitation records and locations to provide broad coverage across British Honduras.

Precipitation data from these 13 stations are summarised in Table 1. The stark contrasts in annual average precipitation between northern Belize (Corozal, Orange Walk, Belize and Cayo Districts) and the southern regions (Stann Creek and Toledo Districts), shown in Table 1 and Fig. 1b, have led to the separate treatment of these areas in this article.

**Table 1**: **Average annual rainfall data for 13 meteorological stations in British Honduras and driest year on record for each station.** * indicates incomplete data over the reference period selected. ** combines rainfall data from the Public Hospital in Belize City (1863-1949), St. John's College in Belize City (1950-1970) and the International Airport (1971-1990) (The British Honduras Colonist and Belize Advertiser, 14 September 1867; The Colonial Guardian, 16 April 1904; Walker, 1973, Rushton, 2014).

| Meteorological Station | Annual Precipitation Average (reference period) | Annual Precipitation in Driest Year on Record |
|---|---|---|
| Santa Rita Corozal | 52.57 inches [1,335.28 mm] (1883-1899) | 33.68 inches [855.47 mm] (1855) |
| Corozal | 59.15 inches [1,502.41 mm] (1903-1967)* | 25.71 inches [653.03 mm] (1923) |
| Corozal Agricultural Station | 54.86 inches [1,393.44 mm] (1936-1970)* | 29.44 inches [747.77 mm] (1949) |
| Orange Walk | 60.11 inches [1,526.79 mm] (1906-1957)* | 33.06 inches [839.72 mm] (1939) |
| Orange Walk Agricultural Station | 53.66 inches [1,362.96 mm] (1944-1965)* | 31.92 inches [810.76 mm] (1959) |
| Yo Creek Agricultural Station | 57.72 inches [1,466.08 mm] (1965-1970) | 38.86 inches [987.04 mm] (1967) |
| Belize City** | 79.07 inches [2,008,37 mm] (1863-1990)* | 41.75 inches [1,060.45 mm] (1923) |
| San Ignacio | 60.64 inches [1,540.25 mm] (1906-1960)* | 31.40 inches [797.56 mm] (1923) |
| Central Farm Agricultural Station | 60.83 inches [1,545.08 mm] (1949-1976)* | 26.09 inches [662.68 mm] (1949) |



| Stann Creek | 88.22 inches [2,240.78 mm] (1906-1959)* | 35.85 inches [910.59 mm] (1959) |
| Stann Creek Agricultural Station | 88.12 inches [2,238,24 mm] (1932-1976)* | 50.60 inches [1,285.24 mm] (1949) |
| Punta Gorda | 164.7 inches [4,183.38 mm] (1895-1968)* | 99.35 inches [2,523.49 mm] (1944) |
| Punta Gorda Agricultural Station | 149.28 inches [3,791.71 mm] (1935-1976)* | 105.94 inches [2,690.87 mm] (1975) |

Data from geographically close stations within the same region have been combined in Fig. 2 and Fig. 3 to provide the longest possible record for seven distinct areas, four in the north and three in the south, each reflecting different precipitation patterns.

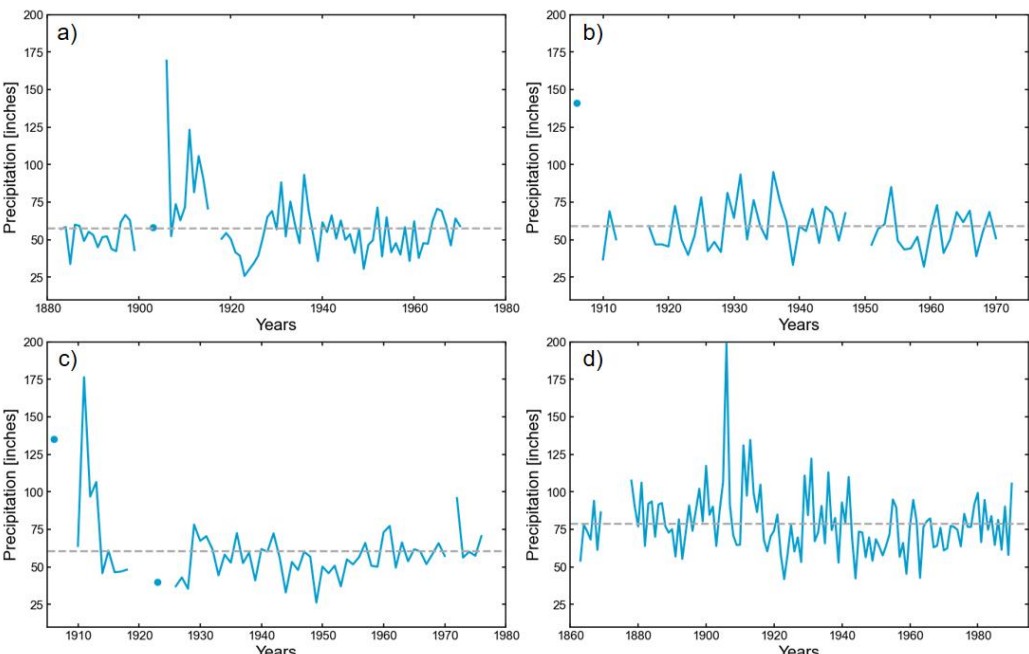

**Figure 2 (a): Annual rainfall totals for Corozal (grey line indicates average in all figures).** Figure combines data from Santa Rita (1883-1899), Corozal (1903; 1906-1915; 1918-1949; 1952-1964; 1967) and Corozal Agricultural Station (1950-1951; 1965-1966; 1968-1970). **(b): Annual rainfall totals for Orange Walk.** Figure combines data from Orange Walk (1906; 1910-1912; 1917-1947; 1951-1957), Orange Walk Agricultural Station (1958-1965) and Yo Creek Agricultural Station (1966-1970). **(c): Annual rainfall totals for San Ignacio.** Figure combines data from San Ignacio (1906; 1910-1918; 1923; 1926-1948; 1951-1960) and Central Farm 135 Agricultural Station (1949-1950; 1961-1970; 1972-1976). **(d): Annual rainfall totals for Belize City.** Figure combines data from Public Hospital (1863-1949), St. John's College (1950-1970) and the International Airport (1971-1990).





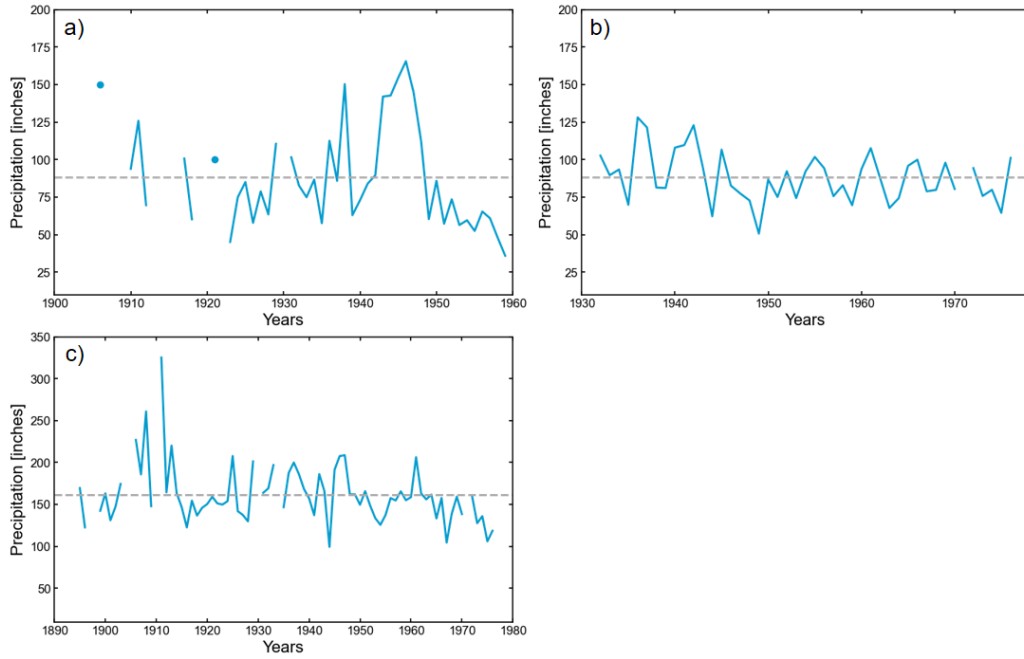

**Figure 3 (a):** **Annual rainfall totals for Stann Creek** (1906; 1910-1912; 1917-1918; 1921; 1923-1929; 1931-1959). **(b):** **Annual rainfall totals for Stann Creek Agricultural Station** (1932-1970; 1972-1976). **(c): Annual rainfall totals for Punta Gorda.** Figure combines data from Punta Gorda (1895-1896; 1899-1903; 1906-1909; 1911-1929; 1931-1933; 1935-1964; 1966-1968) and Punta Gorda Agricultural Station (1965; 1969-1970; 1972-1976).

**3 Historical sources**

The findings presented here result from a rigorous examination of a diverse range of historical sources. These include travel accounts, diaries, private letters, colonial reports, missionary accounts, agricultural, economic and financial reports and rainfall data, published between the late 18th century and the 1980s. Documentary materials were consulted in archives and special collection libraries in Belize and the UK, with the main collections summarised in Table 2. Newspapers published in British Honduras, the USA, the UK and the West Indies from the late 18th century until the 1980s have proved an essential source for the identification of extreme climate events. A total of 288 historical sources documenting droughts in British Honduras were collected. This count excludes instrumental records and duplicate reports published by multiple sources (Fig. 4).

**Table 2**: **Details of main archival collections consulted for this study.**

| Name of repository | Collections consulted and referencing codes |
|---|---|
| Archives of the Society for the Propagation of the Gospel in Foreign Parts, Rhodes House Library, University of Oxford, UK | *E Series of Original Missionary Reports, 1901-1950* (SPG-E) |
| Belize Archives and Records Service, Belmopan, Belize | *Minute Papers* (BARS, MP) |
| Cambridge University Library, Cambridge, UK | *Annual Reports of the Department of Agriculture* |



| National Archives, Kew, London, UK | Colonial Office, sections 123, 127, 128, 137 (CO) |
| --- | --- |

Identifying dry anomalies in pre-20[th] century British Honduras primarily depends on two sources: records of mahogany industry activities and reports on water supplies in present-day Belize City. Drought reconstruction in the 20[th] century is facilitated by the availability of more diverse sources. Notably, the Department of Agriculture's annual reports, along with a growing body of instrumental data, offer extensive insights into rainfall variability across the country.

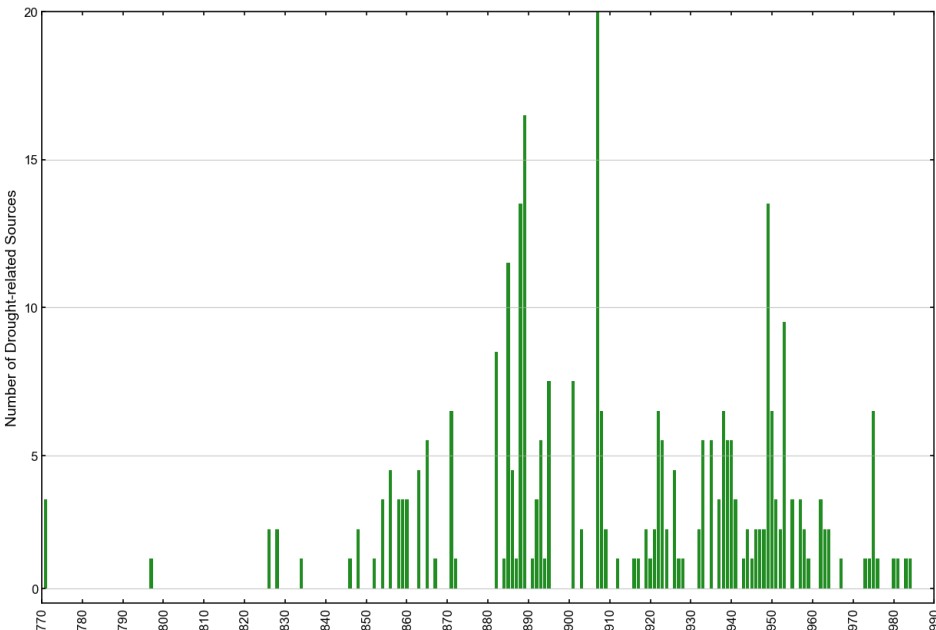

**Figure 4**: **Number of drought-related sources per year across the study area.**

### 3.1 Reports of mahogany industry activities

Since their arrival in the Honduras Bay, British settlers had started to cut and export forest products, mainly logwood (*Haematoxylum campechianum*). Logwood became a valuable commodity in the 17[th] and 18[th] century European textile industry as it was used to produce dyes of several hues. By the mid-18[th] century, as declining prices and foreign competition meant that the logwood industry was no longer profitable, local loggers began to shift rapidly to the extraction of mahogany (*Swietenia macrophylla*), which by the 1770s had become the main export of the settlement (Bolland, 1977; Bulmer-Thomas and Bulmer Thomas, 2012; Rushton, 2014). Mahogany soon became a highly valued decorative hardwood and luxury item in Europe, and saw high exports especially in the periods 1823-1860 and 1900-1930 (Weaver et al., 1997; Rushton, 2014). Although depletion of the mahogany reserves started to alarm colonial authorities in the second half of the 19[th] century and contributed to the slow decline of the forest industry and the gradual rise of agriculture, the mahogany trade remained the primary economic base of the colony at least until the early 20[th] century (Dobson, 1973; Bolland, 1977).



The production of mahogany consisted of three main stages. Felling was usually conducted at the end of the wet season (or soon after Christmas) and around the middle of August, during the 'Mauger season'. Mahogany logs were then hauled to the river banks. Trucking operations were undertaken during the end of the dry season, especially in April and May, when the ground was dry. Mahogany logs were piled along the river banks and tumbled into the water between mid-June and October, when the seasonal rains raised the level of the rivers allowing transportation downstream. Iron chains were fixed at the river mouth to collect the logs, which were then selected by their respective owners and floated to the coastline, ready to be shipped abroad (Henderson, 1809; Gibbs, 1883; Morris, 1883). The mahogany industry in British Honduras was particularly susceptible to climate anomalies, relying on dry soils to facilitate trucking and on floods to transport logs from the interior to the coast. A wet 'dry season' would hinder hauling operations, as trucks would remain stuck in the muddy paths. Dry 'wet seasons', instead, would prevent the formation of floods and therefore impede the floating of the logs downstream, affecting the main export of the colony.

Due to its centrality in the economy of the colony, local newspapers promptly reported the weather effects on the mahogany market. These sources are therefore crucial to track weather anomalies. However, they only provide information on a limited area, as logging operations were usually conducted along the main rivers of the northern part of the country, particularly the Belize, Sibun and New Rivers (Fig. 1a). In addition, the specific area or river affected by the dry anomaly was usually not reported, as sources were mainly written by inhabitants of Belize City waiting for the arrival of the logs. In some cases, these sources do not fully convey the severity of the dry periods, as authors often focused solely on the absence of floods rather than on meteorological conditions. Nevertheless, despite these limitations, accounts of mahogany industry activities remain a critical resource for studying dry anomalies in the colony.

### 3.2 Accounts of Belize water supply

The town of Belize (present-day Belize City) was completely dependent on rainwater for its fresh water supplies until the late 1940s, when the first piped water system was constructed (MacPherson, 2007). Rain falling during the wet season was stored in iron tanks, wooden vats, barrels, tins and every other kind of receptacle possible to survive during the dry season. Due to the town's lack of alternative fresh water sources, inhabitants were often forced to drink well-water during prolonged dry periods, or flee Belize City and relocate inland to rural areas where river water was accessible or to the coastal cays. However, the consumption of well and river water during these times resulted in a rise in cases of diarrhoea and dysentery (Young, 1847; Fisher, 1855; The Belize Advertiser, 8 June 1889; The Times of Central America, 13 March 1896; Evans, 1948). Water supply and the necessity to increase storage and water access to all social strata was the principal concern of both colonial authorities and the local population until the mid-20th century (British Honduras, Medical Department, 1940; Great Britain, Colonial Office, 1959). Belize City, however, was not the only town experiencing water shortages. Water tanks were erected across the entire colony during the 19th and 20th century, and only few settlements, such as Stann Creek Town (present-day Dangriga), were not dependent on rainfall as the locals used river water for their everyday activities (Gibbs, 1883; Avery, 1900; Boyce, 1906). As access to water (both potable and for washing purposes) was limited even during normal dry seasons, drier-than-average periods could have devastating effects on Belize's inhabitants.

Historical records, especially newspapers, meticulously tracked water levels in water storage receptacles, documenting instances when tanks neared or reached emptiness and when emergency supplies were distributed to distressed inhabitants. Despite providing clear information on drought duration and highlighting their effects on the local population, these reports have some inherent limitations. Scarcer-than-average rainfall in the dry season obviously contributed to the



exhaustion of the rainwater supplies. However, water shortages could also be attributed to other factors, including demographic expansion, limitations in water storage capacity and malfunctions in water tanks (The Belize Advertiser, 8 June 1889; The Colonial Guardian, 20 April 1907; Macpherson, 2007). Periods described as particularly dry by the sources, such as the 1880s and the 1920s, coincided with decades of population increase. The population of Belize City nearly doubled within 30 years, increasing from 5,767 inhabitants in 1881 to 10,478 in 1911. Concurrently, the total population of the colony increased from 10,805 in 1881 to 27,145 in 1931 (Evans, 1948). The growing population, often housed in properties without water tanks, increased water demands and strained the already fragile water supply system in place in the colony.

Defective tanks also exacerbated the vulnerability of Belize City's population to droughts. An example occurred in 1907 when, despite the newly installed public tanks having a capacity of 100,000 imperial gallons [454,609 L], only 20,000 [90,921.8 L] were accessible due to significant leakage issues (The Angelus, March 1893; The Colonial Guardian, 7 September 1907). Continuous requests to increase water storage capacity between the 19[th] and the first half of the 20[th] century also show that the available supply was not enough to meet demand. The capacity of public water tanks in Belize City, which provided water to the poorer classes at a nominal cost, increased from 122,000 imperial gallons in 1923 [554,623 L] to 482,599 [2,194,742.6 L] in 1924, before reaching a capacity of 2.3 million imperial gallons [10,455,007 L] with the installation of new tanks in 1936 (CO 123/60, A member of the Late Grand Jury, 22 February 1841; CO 123/62, Public Meeting, 13 November 1841; Great Britain, Parliament, House of Commons, 1910; British Honduras, Medical Department, 1926; British Honduras, Medical Department, 1937). Hence, it is evident that the assessment of drought severity in historical records relied heavily on the balance between water supply and demand, often diverging from purely meteorological criteria. This divergence occurred because the impact of reduced rainfall varied depending on the vulnerability of different towns and social strata. Constrained supply and heightened demand could magnify the local perception of dry periods, potentially distorting the true nature of climate events.

### 3.3 Department of Agriculture annual reports

The dwindling forest resources encouraged the slow development of agriculture from the second half of the 19[th] century. Between the last decades of the 19[th] century and the 1940s, banana, coconut and plantain dominated agricultural exports, while citrus and sugar production started to increase rapidly only from the 1940s and 1960s, respectively (Dobson, 1973). To coordinate the new industry, a Department of Agriculture was created in 1928, which encouraged the formation of Agricultural Stations from 1933, which would function as demonstration centres for local farmers (Wright et al., 1959). In 1935, the Department began issuing annual reports on the colony's agricultural activities, offering detailed descriptions of weather patterns and the impact of rainfall variability and climate anomalies on crops. These reports are the most valuable source for reconstructing drought events and their effects in 20[th] century British Honduras, as they provide comprehensive coverage of climate-related events across the country over five decades. Every report was also accompanied by detailed rainfall and temperature data obtained from the meteorological stations of the country as well as summaries of statistics. The annual reports enable the reconstruction of intra-annual rainfall variability, as they provide detailed weather observations for each month, often correlated with the growth stages of specific crops.

### 4 Methodology





Following standard approaches used in historical climatology (Nash and Endfield, 2002; Nash and Adamson, 2014; Pfister et al., 2018; Nash et al., 2018) the documentary sources have been carefully read with a specific focus on all information related to droughts, dry periods and late seasonal rains. These include both direct data (explicit references to meteorological and hydrological conditions, such as descriptions of droughts and of low river levels) and indirect data (descriptions of the effects of dry anomalies on population and environment, such as famines, food and water shortages, plagues of locusts, damages to crops, forest fires and cattle mortality). The data obtained have been used to prepare a

chronology of droughts in Belize from the late 18th century until 1981. When possible, instrumental data were used to verify the drought reconstruction and to integrate documentary sources when incomplete or too general. In particular, when the historical records reported weather anomalies with a low degree of spatial or temporal precision, rainfall data were used to identify the area affected by droughts and their duration. In some cases, it has not been possible to identify when exactly these events happened, and therefore only the year of occurrence has been indicated.

Each drought was grouped into one of three severity categories, adapted from Garnier's (2018) indexed scale of drought severity: 'exceptional drought'; 'drought'; 'dry period'. Exceptional droughts are defined as unusually long and continued events that caused remarkable water shortages, famine or scarcity of provisions, unprecedentedly low river levels, crop losses, cattle mortality, extensive forest fires and sometimes forced authorities to obtain additional water supplies. Years with record low rainfall are also in this category. Events in the drought category can be similar in terms of duration but

tend to have higher precipitation and register fewer impacts than the exceptional droughts. This category also includes events described in documentary records as prolonged droughts, even though they may not be reflected as such in the rainfall data. Finally, dry periods indicate unexpected dry anomalies, late seasonal rains, lack of seasonal floods and drier-than-average spells that did not cause serious agricultural, economic or environmental damages.

Following Kelso and Vogel (2007), each event was assigned a confidence rating (CR) from 1 (low) to 3 (high). A rating
of 1 was awarded when only a single source reported the event or when the description lacked detail. In contrast, a rating of 3 was assigned to events corroborated by multiple sources, with precise details on dates, locations and impacts. Events documented by reliable sources, such as the Department of Agriculture annual reports, and supported by instrumental data, also received a rating of 3 (Fig. 7).

**5 Results**

**5.1 Historical records of droughts in the north (18th-20th centuries)**

Figure 5 shows the documentary-derived drought index for the northern half of British Honduras (Corozal, Orange Walk, Belize and Cayo districts, Fig. 1a) covering the period from the late 18th century to the early 1980s. The earliest documented event dates back to 1771, when a drought-induced locust plague triggered a famine in the Bay of Honduras,

particularly affecting "the Indians in the back part of the country … by which means provisions of all kinds were so scarce in the Bay". At Ambergris Caye, the largest island of Belize, 17,000 locals were said to have died for want (Pennsylvania Gazette, 24 September 1771; New York Gazette, 28 October, 1771). In June 1797, it was reported that "an uncommon drought had destroyed all the plantations", from which emerged a famine that left the settlement of Belize with no provisions, forcing the local Superintendent to ask Jamaica for food supplies (CO 137/98, Barrow to Balcarres, 16 June

1797).



References to severe dry periods appear between the mid-1850s and the early 1870s. In 1854, "the fortunate and extraordinary continuation of the dry weather" was beneficial to the mahogany operations and allowed the trucking out of about 8 million feet of wood [226,534.4 m³] (The Packet Intelligencer, 17 June 1854). Only 2 years later, sources reported an early and "very long dry season" that started in January and was accompanied by "an insufferably hot and

dry" weather. Accounts from Belize City reported that by the end of March, "our tanks are all empty – our streets are as dry and dusty as they should be in June, when our dry season will be over, and our pasturage is all burnt up; while our cattle are dying for want of food in the pastures near town" (New York Weekly Herald, 19 April 1856; New York Herald, 15 July 1856). A well-documented event occurred in 1865, when an "unusually long and severe" drought dried up almost every water tank and even the swamps around Belize City. The drought started in February and lasted until early-June,

causing such a severe water shortage that plans were discussed to supply Belize City with water from the Sibun River (The Colonist, 11 February 1865; The Colonist, 29 April 1865; The Colonist, 6 May 1865; The Colonist, 3 June 1865).

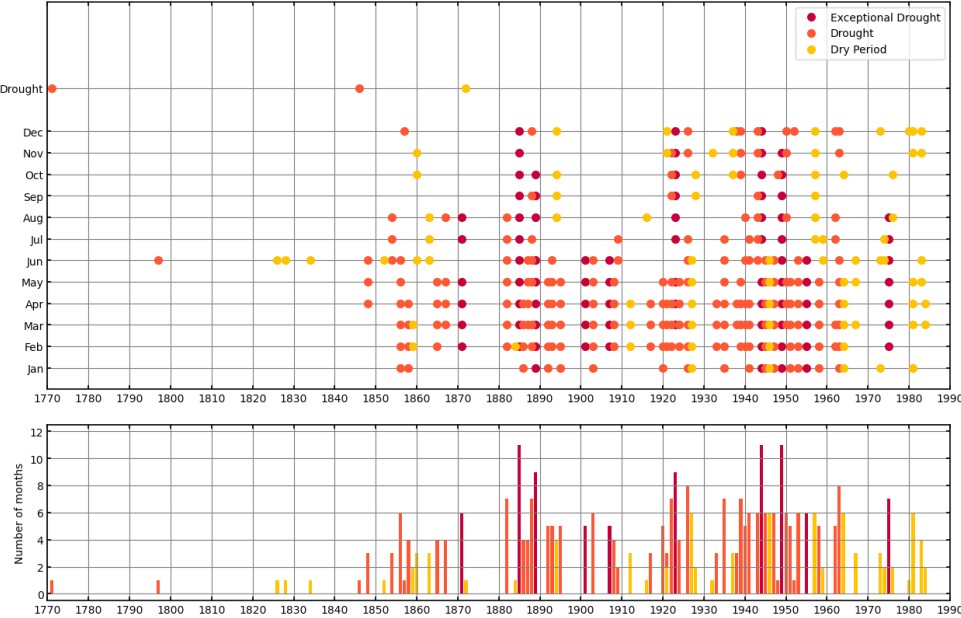

**Figure 5**: **Documentary-derived reconstruction of droughts in the northern half of Belize.** The top plot displays the chronology
of droughts, indicating the specific months in which they occurred, while the bottom plot shows the total duration of each drought in months per year.

The longest drought reported in the period occurred in 1871 (Fig. 5). A "long and almost unexampled period of dry weather with oppressive tropical heat" lasted from February until at least early September, even though some sources

reported that it continued well into the 1872 dry season. Water tanks in Belize City once again proved insufficient to meet local demands and considerable suffering was experienced, as water became "as precious as liquid gold". Local churches, especially St. Mary's, distributed water to the poorer people during the severe drought. The continued dry spell also affected the young sugar canes and prevented the ratoons (new shoots that grow from near the root of crop plants after the old growth has been cut back) from increasing in bulk (New Era and British Honduras Chronicle, 6 May 1871; New





Era and British Honduras Chronicle, 27 May 1871; New Era and British Honduras Chronicle, 16 September 1871; New Era and British Honduras Chronicle, 2 December 1871; New Era and British Honduras Chronicle, 11 December 1871; New Era and British Honduras Chronicle, 18 May 1872).

Evidence suggests that the 1880s were characterised by numerous and unprecedentedly severe dry anomalies. The northern half of the colony experienced a "great drought" in 1882 that started in February and continued at least until
August. The drought particularly affected the northernmost areas, present-day Corozal district, causing the failure of the maize crop and the subsequent misery among the farming communities. The unusual dry weather prevented the seasonal floods at the start of the wet season and therefore "acted so injuriously on the mahogany trade", while proving beneficial to sugar planters. It also led to a dramatic reduction in plantain exports by more than 80 % (The Colonial Guardian, 2 September 1882; The Belize Advertiser, 30 December 1882; British Honduras, Colonial Secretary's Office, 1883; British
Honduras, Colonial Secretary's Office, 1884). In 1885, an exceptional drought hit the northern districts, "so severe as to have withered up the corn [maize] crop, and greatly retarded the growth of canes". The drought destroyed the sugar cane harvest at Corozal, and only a few stumps of the cane were left protruding from the ground. It was an exceptionally long event that lasted from February to December and that brought "the aboriginal population to the brink of famine". Serious famine was avoided only thanks to the "provident habits of the Indians in always storing half their crop every year to
meet such contingencies". Locusts, which had invaded the northern half of the colony the year before, appeared once again in the north, though no significant damage was reported. The drought was so prolonged that it dried out forest paths enough to allow mahogany trucking operations to take place in October, an unusual occurrence (The Colonial Guardian, 15 August 1885; The Angelus, August 1885; The observer, 17 October 1885; The Colonial Guardian, 19 December 1885; The Colonial Guardian, 2 January 1886).

The second half of the 1880s saw severe dry spells in succession. In the early months of 1886, a "long continued drought" in the Belize and Corozal districts brought serious scarcity of water, "locked up wood in the bush" by hindering its transportation and caused extensive fires that burnt even the rushes of the swamps surrounding Belize City (The Colonial Guardian, 1 May 1886; The Belize Advertiser, 25 December 1886). The following year, a long dry season dried-up most of Belize City water supplies by early-May (The Colonial Guardian, 7 May 1887) while in 1888 a "very unusual" dry
season caused water shortages, an increase in the number of cases of dysentery and the lack of floods in October, "a fact unprecedented and beyond the recollection of the oldest inhabitant" (Lucas, 1890; The Belize Advertiser, 3 November 1888). The drought continued until the second half of 1889. By the end of May, water was extremely scarce in Belize City, and most of the poorer inhabitants left the town for the interior. By June, the event was described as a "lengthened period of extreme drought" that had dried up the ponds in the country districts and that caused river water to be brackish.
The Belize River registered the lowest water level of the past 20 years, as "water at the usual 7 ft [2.1 m] depth cannot be found on the ridges". Despite a rainy July, the drought re-appeared in September, when Belize City inhabitants were "as badly off for water as they were in the dry season". The abundant rains of November finally broke the long dry spell (The Belize Advertiser, 25 May 1889; The Belize Advertiser, 22 June 1889; The Belize Advertiser, 6 July 1889; The Belize Advertiser, 28 September 1889; The Belize Advertiser, 2 November 1889).

Evidence shows that the 1900s were characterised by severe dry events. In 1901, a dry anomaly affected the north of the colony (Fig. 5), "so prolonged that no one old enough exists in Belize who can remember any other of such duration". By April, the dry weather had arrested the growth of the grass and the lack of water was becoming severe. Water tanks were probably empty by May, as Belize City authorities had to send lighters to the Quamina Creek to bring potable water into town (The Colonial Guardian, 6 April 1901; The Colonial Guardian, 25 May 1901; The Colonial Guardian, 22 June 1901;



The Colonial Guardian, 30 May 1908). The "long drought" that hit the colony in 1903 reducing the banana production did not cause the same water shortages as the 1901 drought, due to increased water supply (The Colonial Guardian, 25 April 1903).

In 1907, however, the colony was once again in the grip of an "exceptional drought", "the longest known to anyone now living in Belize" which "exceeded that of 1901 and was unprecedented within the memory of the oldest inhabitant". As Belize City's water supply ran short in April, drinking water again had to be fetched from Quamina Creek. As this was insufficient, water was also sent from New Orleans and Mobile (Louisiana, USA). The lack of rain made planting impossible in the north and ruined the maize crop while also reducing the gum produced by rubber trees. By June, the heat had become so "frightful" that a part of the population suggested offering public prayers, as temperatures in the Belize and Corozal districts had reached 97 °F [36.6 °C]. Low water levels in the main rivers of the north further indicated the severity of the drought. At the height of the drought, no motor boats could reach San Ignacio due to the rivers drying up, and no lighters could navigate upstream across the Sibun River bar. The rains of early July finally broke the drought (The Colonial Guardian, 27 April 1907; The Colonial Guardian, 11 May, 1907; The Colonial Guardian, 22 June, 1907; The Colonial Guardian, 2 November 1907; The Colonial Guardian, 30 May 1908; Great Britain, Parliament, House of Commons, 1908; CO 128/88, British Honduras Blue Book for 1907).

A new period marked by numerous dry events emerged in the early 1920s (Fig. 5). "Exceptionally dry weather" experienced in 1919 and 1920 caused a decrease in the coconut exports (Great Britain, Parliament, House of Commons, 1922), while in 1921 severe drought affected the Belize district, forcing the authorities to transport water from the Manatee River (The Time, 23 June 1921). In 1922, the "failure of rains" caused the almost complete ruin of chicle production—a natural latex obtained from the sap of the sapodilla tree (*Manilkara zapota*) and historically used as a primary ingredient in chewing gum—as well as a significant decline in the maize crop in Corozal. Belize City again had to fetch potable water from Young's Pond, on the Belize River. The drought continued up to the end of the year causing a great economic depression that, coupled with low sugar prices, triggered the exodus of traders and artisans to Mexico. The year was also very dry in the Cayo district, where trucking of mahogany logs was done in November, "a thing unknown for years" (SPG, E-1922, Report of Reverend J. M. Shaw, 31 December 1922; BARS, MP, Box 130, 613, Annual Report on Corozal District for 1922; BARS, MP, Box 130, 648, Annual Report Cayo District for 1922; British Honduras, Medical Department, 1923). The next year (1923), a prolonged drought started around March and lasted until the end of the year, hampering the mahogany and chicle operations. In Orange Walk, the water of the New River was reported to be "lower than it has ever been". Due to defective tanks, Belize City had to transport additional water supply from Tillet's Pond and sell it to the public. The extremely dry year was considered the cause of the extremely high infantile mortality registered in the Cayo district (Great Britain, Parliament, House of Commons, 1924; British Honduras, Medical Department, 1924; Walker, 1973). In 1924, Belize City water supplies had to be replenished once again with freshwater coming from Young's Pond (British Honduras, Medical Department, 1926).

An "abnormally severe and prolonged dry season" affected large parts of the colony between January and July 1935. The maize, banana and rice crops were severely damaged by the dry spell, and many fields were re-seeded three times before growth was established. As many planters found themselves short of seed, it was necessary to distribute free seeds to 650 planters in the Belize and Cayo districts. The drought also affected the livestock farming industry, as some farmers lost "some of their stock through a lack of suitable food and water" (British Honduras, Department of Agriculture, 1936; British Honduras, Medical Department, 1936).



The colony experienced some of the most severe droughts in its history between 1943 and 1955 (Fig. 5). In 1943, the
weather was "abnormal", as "the dry season was very wet and the wet season was dry". The drought reduced the maize
crop in the north by about 50 % and rice suffered in all parts of the colony due to the lack of rain (British Honduras,
Department of Agriculture, 1944). The dry spell persisted until the end of 1944, which was defined "an exceptionally dry
year; the driest, in fact, since rainfall records have been kept". "Unduly dry weather" in May made replanting of maize
necessary in many areas, and the drought caused a reduction of the sugar cane crop in the north. Due to the prolonged
drought in Belize City, water in the vats and tanks was exhausted and additional supply was obtained from the Belize
River at a point located 20 miles [32.1 km] away from the town (British Honduras, Department of Agriculture, 1945;
British Honduras, Medical Department, 1945). Prolonged dry seasons in 1946 and 1947 made conditions difficult for the
establishment of the maize crop and affected the livestock industry, causing the death of many animals in the northern
districts and in Cayo (British Honduras, Department of Agriculture, 1947; Great Britain, Colonial Office, 1948; British
Honduras, Department of Agriculture, 1948).

During the second half of 1948, a "period of exceptional drought" reduced the yields of all crops and largely destroyed
the winter vegetable crop (British Honduras, Department of Agriculture, 1949). It was the prelude to "one of the most
severe droughts experienced in the history of the colony" that lasted from the end of 1948 until December 1949. The dry
conditions hit large parts of British Honduras and, coupled up with the decline of the mahogany and chicle industries,
caused an economic crisis and increasing unemployment especially in the north. The lack of rain caused the almost
complete failure of the many successive replantings. By the end of the year, reserves of grain and seeds were considerably
reduced and, in many cases, exhausted. Around 3,500 head of cattle died during the year, and food surpluses produced
by the wetter southern regions had to be sent to relieve the drought-stricken northern districts. The drought triggered fierce
bush fires that devastated 250 square miles [64,750 ha] of hardwood chicle forest, stripped the undergrowth and smaller
trees off the hillsides, and baked the exposed ground brick-hard. The year was difficult for the mahogany industry, as the
first river floods appeared only in December, and no logs could be transported on the Belize River until January 1950.
The drought of 1949 caused a shortage of food crops that continued well into 1950. The authorities resolved to the
importation of relief maize, which was ended only late into 1950 with the appearance in the market of the 1950 maize
crop (Great Britain, Colonial Office, 1950; British Honduras, Department of Agriculture, 1950; The Times, 29 December
1949; Daily Mail, 10 February 1950; British Honduras, Public Relations Office, February 1950; CO 127/45, Government
Gazette Extraordinary, 29 December 1950; BARS, MP, Box 202, 417, Annual Report Cayo District for 1950, 12 March
1951).

Dry conditions re-appeared in early 1950, as the colony was recovering, with difficulty, from the exceptional drought of
the year before. Acute shortage of water was felt especially in the Corozal and Orange Walk districts, where new water
cisterns were erected and wells drilled. Livestock owners and farmers suffered from lack of food for animals, and fires
were again extensive although not as prevalent as in 1949 (British Honduras, Public Relations Office, May, June, August
1950; CO 127/45, Government Gazette Extraordinary, 29 December 1950; BARS, MP, Box 202, 417, Annual Report
Cayo District for 1950, 12 March 1951). Early and severe dry spells hit the colony in 1951 and again in 1953, both leading
to extensive bush fires, especially in the latter year (British Honduras, Department of Agriculture, 1952; BARS, MP, Box
214, 329, Annual Report Forest Department for 1953; British Honduras, Department of Agriculture, 1954). The final
exceptional event in this period occurred in 1955, with "an intense dry that rivalled that already notorious in 1949", lasting
from January to June. The drought brought some of the "worst forest fires for many years" and extremely high
temperatures between April and June, reaching over 100 °F [37.7 °C] in Baking Pot, northeast of San Ignacio. The





estimated value of timber lost was approximately £714,000. Of 450 square miles [116,550 ha] of pinelands in the colony,

about 300 square miles [77,700 ha] were affected. The lack of rain delayed planting and several replantings were needed. It also caused considerable hardship for cattle, and losses were serious, though not as much as in previous years (The Daily Telegraph, 28 May 1955; Great Britain, Colonial Office, 1957; British Honduras, Department of Agriculture, 1956).

The last exceptional drought in colonial Belize occurred in 1975, with extremely dry conditions persisting from February to early September. The dry conditions reduced maize production by 42 %, and seed shipments arriving from Mexico

were necessary to relieve local farmers. Raw sugar production was also reduced by 25 % and oranges by 27 %. The livestock industry was hit hard by the drought. By the end of July, an estimated 200 head of cattle had died, and the number rose to 525 in September. Heavy showers in September and October ended the long drought, but its economic effects extended into 1976 (British Honduras, Department of Agriculture, 1976; Financial Times, 8 June 1976; British Honduras, Department of Agriculture, 1977; Financial Times, 21 January 1977; Hall, 1983).

**5.2 Historical records of droughts in the south (19ᵗʰ-20ᵗʰ centuries)**

Figure 6 shows the reconstruction of dry anomalies in the southern half of Belize (the districts of Stann Creek and Toledo, Fig. 1a). Descriptions of dry events in the south are usually scant and only rarely report the effects caused by dry anomalies. The first references to dry periods appeared at the end of the 19ᵗʰ century. Sources report that the Toledo district experienced "the driest season for years" in April-May 1892, when several sugar cane fields caught fire due to the

"exceedingly dry" conditions. In the same period, the northern districts were experiencing an "unusually dry season" lasting from January to May (The Colonial Guardian, 14 May 1892; The Colonial Guardian, 11 June 1892). In early 1895, an "exceptionally long spell of dry weather" caused considerable friction over water supplies in the Stann Creek district, while the northern districts were also severely affected by the drought (The Colonial Guardian, 27 April, 1895; The Colonial Guardian, 11 May 1895). Dry anomalies seem to intensify between the late 1910s and the mid-1920s. In 1919,

the Toledo district had a "long and excessively dry" start of the year, particularly between January and May (British Honduras, Medical Department, 1922). The exceptionally dry 1923 (see above) also affected the southern areas. The year "was an exceptionally dry one with about half the average precipitation" in the Stann Creek district and the seasonal floods did not occur. 1923 marked a period of "particularly severe" drought also in Punta Gorda, prompting authorities to implement water rationing measures in public vats to cope with the prolonged dry conditions (Stann Creek Railway,

1924; British Honduras, Medical Department, 1924). Three years later, the Stann Creek district again experienced an "exceptionally dry [year] and so the Stann Creek River did not rise as often as usual" (BARS, MP, Box 175, 1183, Annual Report on the Stann Creek District for the year 1926, 27ᵗʰ April 1927).




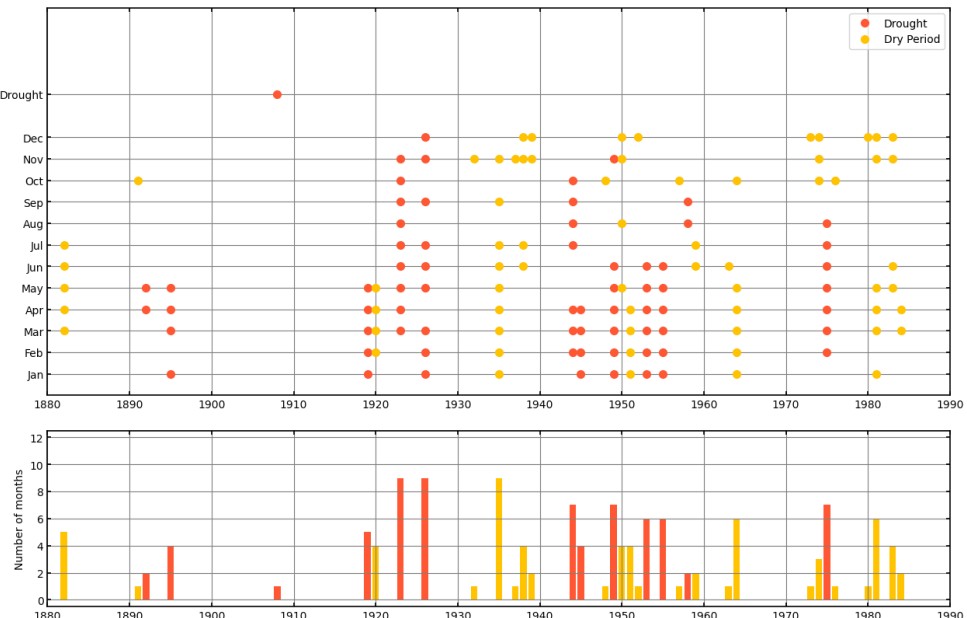

**Figure 6**: **Documentary-derived reconstruction of droughts in the southern half of Belize.** The top plot displays the chronology of droughts, indicating the specific months in which they occurred, while the bottom plot shows the total duration of each drought in months per year.

Between 1945 and 1955, the southern areas experienced several abnormally dry periods. In 1945, the Stann Creek district experienced "unusually severe" dry weather in the first four months of the year, registering the lowest rainfall since 1935 (British Honduras, Department of Agriculture, 1946). Drought conditions reemerged in 1949. While the northern districts were severely impacted, conditions in the south "were closer to normal". However, when requests were made to send supplies to the north, only "limited surpluses of food crops" were available in the southern areas (British Honduras, Department of Agriculture, 1950). The south suffered from a new long drought in 1953. In Toledo district, the year was comparatively dry and both animals and pastures lost condition in the first half of the year. Abnormally dry conditions also prevailed in the Stann Creek district, especially during the first six months, where plantain production was badly affected by the dry spell. By March, pastures had dried up, livestock was falling off in condition and extensive fires had burnt large areas of pine woodlands south of Stann Creek (British Honduras, Department of Agriculture, 1954; British Honduras, Public Relations Office, March 1953; BARS, MP, Box 214, 397, Annual Report on the Toledo District for 1953; BARS, MP, Box 214, 329, Annual Report Forest Department for 1953). In 1955, the Toledo district "had been in trouble during the early and prolonged dry" that hit the south in the first six months, damaging crops and contributing to what was called "the worst season of the century" (British Honduras, Department of Agriculture, 1956). Finally, the exceptional drought that struck the entire country in 1975 also caused a prolonged dry spell in the Stann Creek district, lasting from February to September (British Honduras, Department of Agriculture, 1976).

**5.3 Comparison between the north and south**

The documentary-derived reconstruction of droughts shows that droughts in the south were far less numerous and intense than in the north. It also reveals that abnormally dry periods impacted the northern half of the colony during the years



1882-1889, 1901-1907, 1920-1924 and 1943-1955, with severe and isolated events in 1935 and 1975. The southern areas experienced drier-than-average periods from 1919 to 1926 and from 1945 to 1955. Isolated dry spells also occurred in
1895 and 1975. The major dry events that occurred in the early 1920s and again between the mid-1940s and mid-1950s affected the entire colony, albeit with varying intensity. The northern districts experienced the driest climatic conditions during these periods. The droughts of 1935 and 1975 were also felt across large parts of Belize, although damage in the south was not serious. The severe dry events that hit the north in the 1880s do not appear to have affected the southern districts significantly, as only one dry event is recorded in that decade. In 1885, while the northern districts were suffering
from a long drought, the Toledo district experienced four months of excessive rains from June to September (The Colonial Guardian, 11 October 1885). Similarly, the exceptional droughts of 1901 and 1907 did not affect the southern districts. The only recorded dry event of the 1900s in the south occurred in 1908, when fires affected the Stann Creek district.

Dry events in the south rarely went unrecorded in the north. However, in October 1891, the Toledo district experienced an "almost entirely dry" month, with no corresponding dry events reported in the north (The Colonial Guardian, 14
November 1891). In 1919, the same district experienced excessively dry weather, with January, March, and May drier in Punta Gorda than in Corozal (2 inches [50.8 mm], 0.78 inches [19.81 mm] and 2.25 inches [57.15 mm] compared to 2.60 inches [66.04 mm], 1.29 inches [32.77 mm] and 4.56 inches [115.82 mm], respectively). Finally, in October 1974, the country experienced abnormal precipitation patterns, with Corozal receiving three times more rainfall than the Toledo District (18.62 inches [472.55 mm] compared to 5.35 inches [135.89 mm]) (British Honduras. Department of Agriculture,
1975).

Ten exceptional events were identified in the north (1871, 1885, 1889, 1901, 1907, 1923, 1944, 1949, 1955 and 1975). In most of the cases, droughts covered the first half of the year (the dry season) and ended in early June with the arrival of the seasonal rains. However, on seven occasions (1871, 1885, 1889, 1923, 1944, 1949 and 1975) the drought extended well into the second half of the year. Exceptional droughts were usually the longest events recorded. Three droughts lasted
at least 11 months (1885, 1944 and 1949), making them the longest recorded. In contrast, the shortest exceptional events lasted five months (1901 and 1907). In the north, seven events classified as 'droughts' lasted seven or more months (1882, 1888, 1922, 1926, 1935, 1939 and 1963). Although these events lasted longer than some of the exceptional droughts identified, they were reported to be less severe and caused less damage. In the south, no exceptional occurrences were identified, whereas 13 events were classified as 'droughts'. Their durations spanned from two to nine months. The longest
dry spells were recorded in 1923, 1926 and 1935. Nevertheless, the 1935 event has been categorized as a 'dry period' due to the absence of documentary records referencing such a prolonged dry spell.

## 6 Discussion

### 6.1 Comparison of documentary and instrumental data

This section compares Fig. 2-3 with Fig. 5-6 to highlight the degree of consistency between the documentary-derived reconstruction and the instrumental data. The incorporation of confidence ratings in the documentary-derived reconstruction of droughts helps mitigate the inherent biases and limitations of historical records. Of the 83 droughts identified across the whole country, 50 were assigned a CR 3 (60,2 %), 21 were classified as CR 2 (25,3 %) and 12 received a CR 1 (14,5 %) (Fig. 7).



The comparison of documentary and instrumental data reveals that the significant dry period affecting the north from 1882 to 1889, as indicated by historical sources, is partially reflected in the precipitation records from the two active stations at the time: the Public Hospital in Belize City and Santa Rita Corozal. Belize City registered a below-average value of 63.89 inches [1,622.53 mm] in 1882, confirming the incidence of a drought that year (see Table 1 for comparison with annual averages). However, the values recorded in 1885 and 1889 (69.91 inches [1,775.09 mm] and 72.74 inches
[1,846.36 mm] respectively) do not seem to corroborate the exceptional droughts reported by historical sources in those years, both of which received a CR 3. In 1885, Santa Rita recorded one of its lowest precipitation levels on record, totalling just 33.68 inches [855.47 mm], compared to the average of 52.57 inches [1,335.28 mm]. The annual precipitation record for 1889 does not reflect the drought reported by documentary sources, as Santa Rita registered 55.19 inches [1,402.83 mm], a value above the average. However, the first five months of the year were dry, particularly January, which recorded
a rainfall of 0.76 inches [19.3 mm] compared to an average of 2.43 inches [61.72 mm]. The data also confirm that the drought persisted into the latter part of the year, with October recording just 3.96 inches [100.58 mm], well below the average of 6.85 inches [174 mm].

Annual precipitation totals remained below average in 1888 and from 1891 to 1895, correlating with the emergence of dry anomalies observed in the documentary-based reconstruction. Belize City and Corozal stations do not report
particularly low figures in the decade from 1900 to 1910. In particular, the years 1901 and 1907, characterised by exceptional droughts according to the documentary sources (both assigned a CR 3), recorded above-average values in Belize City (84.62 inches [2,148.33 mm] and 91.17 inches [2,316.62 mm] respectively). Despite being described as exceptionally prolonged, instrumental data indicate that the 1901 drought only lasted from February to May. In Corozal, 1907 was below normal (52.2 inches [1,327.08 mm]).


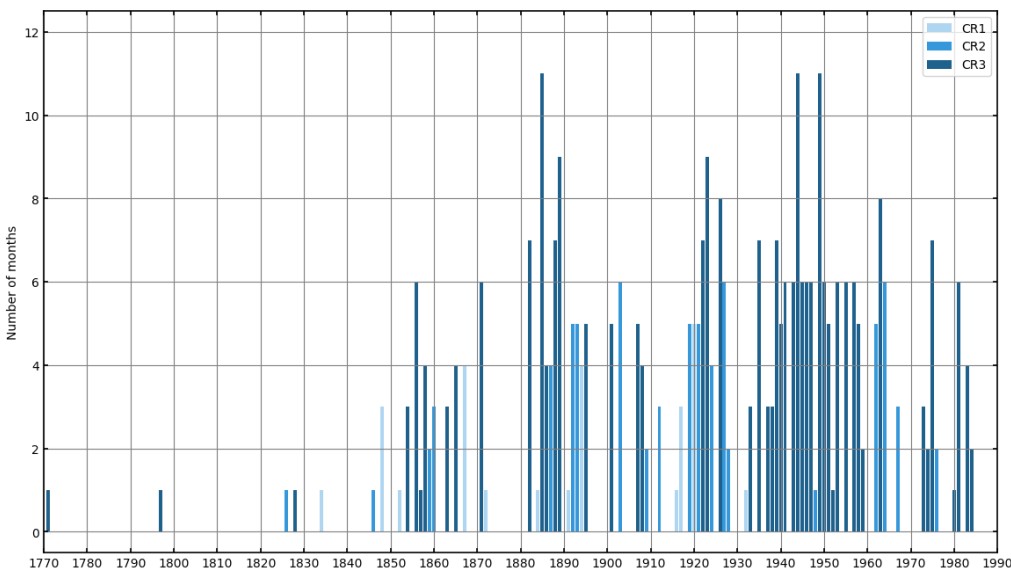

**Figure 7**: **Confidence ratings for identified dry events. Based on Kelso and Vogel (2007).**



The four stations of the north (Corozal, Belize City, Orange Walk and San Ignacio) present slightly different data for the 1910s. Belize City recorded extraordinarily wet conditions between 1911 and 1916, particularly in 1911 (130.93 inches [3,326.7 mm]), 1913 (134.56 inches [3,419.61 mm]) and 1916 (104.74 inches [2,661.35 mm]). Corozal registered abnormal rainfalls between 1911 and 1914, especially in 1911 (123.21 inches [3,132.93 mm]) and 1913 (105.64 inches [2,684.48 mm]). In San Ignacio and particularly in Orange Walk, the wet period was shorter. In contrast, all four stations report a significant dry period between the late 1910s and the late-1920s, corroborating the drought reconstruction. In particular, 1923 (assigned a CR3) was the driest year on record for Belize City (41.75 inches [1,060.45 mm]), Corozal (25.71 inches [653.03 mm]), and San Ignacio (31.40 inches [797.56 mm]) and one of the driest on record for Orange Walk (39.71 inches [1,007.63 mm]) (see Table 1 for comparison with annual averages). All four stations reported dry years in 1935 and 1939, further corroborating the drought reconstruction, which identified seven-month-long droughts on both occasions. The lowest amount in 1935 was recorded at Corozal, with 47.58 inches [1,208.29 mm], while in 1939, Orange Walk registered the record low with 33.06 inches [839.72 mm], compared to an average of 60.11 inches [1,526.79 mm]. Both years were assigned a CR 3 for the historical records.

The significant dry period that affected the north from 1943 to 1955 is clearly reflected in the records from Belize City, Corozal and San Ignacio, which consistently reported consecutive years of below-average precipitation. The longest of these dry spells was recorded in San Ignacio, where drier-than-average years occurred between 1943 and 1956. Conversely, Orange Walk experienced a decade marked by more variable conditions. The exceptional drought identified by sources in 1944 (CR 3) is confirmed by the rainfall data from Belize City and San Ignacio. Both areas endured one of the driest years on record, with rainfall measuring just 42.20 inches [1,073.08 mm] and 32.88 inches [834.91 mm], respectively. However, in Corozal and Orange Walk, the year was wetter than usual, with 62.75 inches [1,591.85 mm] and 71.84 inches [1,822.78 mm] of rainfall, respectively (see Table 1). In contrast, all the available data for 1949 confirm the occurrence of an exceptional drought in the north, notably in Central farm Agricultural Station, where the driest year on record was documented with just 26.09 inches [662.68 mm], and in Corozal, with 30.47 inches [774.98 mm].

Data from the second half on the 1950s show some discrepancies with the drought reconstruction. Only rainfall records from Corozal and Orange Walk substantiate the severe drought of 1955 (historical records CR 3), while in Belize City the year as a whole was wetter than average, with 94.86 inches [2,410.16 mm]. Instrumental data suggest that the end of the decade was drier than reported by documentary sources. In particular, 1959 (CR 3) was the driest year on record for Orange Walk and a very dry one also for Belize City and Corozal, with 31.92 inches [810.76 mm], 45.24 inches [1,149.70 mm] and 35.73 inches [907.58 mm], respectively. However, documentary sources only reported "comparatively dry weather in June and July" (British Honduras, Department of Agriculture, 1960). Finally, the limited data from the north corroborate the existence of the 1975 drought, especially evident in Belize City, where rainfall totalled only 8.53 inches [216.66 mm] between February and early September, well below the average of 35.94 inches [912.87 mm] for the same period.

In the southern half of the colony, rainfall data from Stann Creek and Punta Gorda suggest that the significant dry period, which documentary sources indicate lasted from 1919 to 1926, might have begun earlier. Stann Creek reported below-average annual totals in 1912 (69.59 inches [1,769.02 mm]) and 1918 (60.15 inches [1,527.81 mm]), followed by a prolonged period of consecutive dry years from 1923 to 1928. Punta Gorda, instead, registered ten consecutive drier-than-average years between 1915 and 1924, with notably low precipitation in 1916 (122.55 inches [3,112.77 mm]) and 1918 (136.62 inches [3,467.79 mm]). These two years were not identified as drought periods in the historical records consulted. Instrumental data from Punta Gorda confirm an abnormally dry start to 1919, with January recording just 2 inches [50.8



mm] of rain, significantly lower than the average of 9.1 inches [231.14 mm]. May experienced similarly dry conditions, registering 2.25 inches [57.15 mm], well below the average of 10.45 inches [265.43 mm] (Walker, 1973). Precipitation data confirm the prolonged droughts recorded in historical records in 1923 (CR 3) and 1926 (CR 3), with Stann Creek recording only 45.07 inches [1,143.78 mm] and 57.80 inches [1,468.12 mm] of rainfall, respectively, compared to an average of 88.22 inches [2,240.78 mm]. However, Punta Gorda station registered two consecutive drier-than-average years in 1927 (137.40 inches [3,488.76 mm]) and 1928 (129.80 inches [3,295.72 mm]), the driest since 1916, which are not reflected in the documentary reconstruction.

The significant dry period experienced in the south between 1945 and 1955 is only partially evident from the rainfall data available. Instrumental records also suggest that droughts may have begun earlier in some areas. Stann Creek registered an exceptionally wet period between 1943 and 1948, with particularly high rainfall in 1945 (154.57 inches [3,925.72 mm]) and 1946 (165.52 inches [4,203.21 mm]), apparently contradicting the documentary sources. The nearby Agricultural Station, instead, registered below-average annual precipitation in 1944 (62.14 inches [1,578,56 mm]) and between 1946 and 1951, with lowest precipitation on record in 1949 (50.60 inches [1,285.24 mm]). Precipitation data reveal that 1944 (CR 3) was extremely dry in the Toledo district. Punta Gorda registered its driest year on record, with only 99.35 inches [2,523.49 mm], significantly below the average of 164.7 inches [4,183.38 mm]. Notably, no documentary records mention dry conditions in the south. Conversely, the year was extremely wet in Stann Creek, with 142.65 inches [3,620.61 mm], compared to an average of 88.22 inches [2,240.78 mm]. Starting in the late 1940s, active meteorological stations reported distinctly different precipitation patterns. Stann Creek experienced below-average precipitation between 1949 and 1959, culminating in 1959 with the lowest recorded precipitation of 35.85 inches [910.59 mm]. Conversely, Stann Creek Agricultural Station did not report prolonged dry periods but rather experienced more variable conditions until the mid-1970s. Punta Gorda, instead, experienced below average annual precipitation between 1952 and 1976, particularly in 1954 (125.66 inches [3,194.82 mm]) and 1967 (104.48 inches [2,652.35 mm]), with the exception of a wetter period from 1958 to 1962, peaking in 1961 (206.30 inches [5,237.82 mm]). No dry events were reported in 1954, while 1967 was assigned a CR 2. Finally, data from the Stann Creek Agricultural Station and Punta Gorda Agricultural Station confirm the severity of the 1975 drought (CR 3), with recorded rainfall amounts of just 64.47 inches [1,637.92 mm] and 105.94 inches [2,690.87 mm], respectively (see Table 1).

## 6.2 Comparison with regional drought records

The results presented in Fig. 5-6 can be compared with existing documentary-derived drought chronologies for other Mexican, Central American and Caribbean regions. Table 3 lists the major historical droughts in the Mexican Yucatan Peninsula, Antigua, Guatemala City and British Honduras from the mid-18th century. Figure 8 provides a visual representation of these occurrences.

**Table 3**: **Historical droughts in the Mexican Yucatan Peninsula, the island of Antigua, Guatemala City and British Honduras.** Years highlighted in bold denote periods of drought also experienced in British Honduras (Florescano and Swan, 1995; Mendoza et al., 2007; Márdero et al., 2012; Berland et al., 2013; Guevara-Murua et al., 2018).

| Place | Years of historical droughts |
| --- | --- |
| Mexican Yucatan Peninsula (period covered: 1765-1980) | **1765-1774**; 1800-1805; 1807; 1809-1810; 1813; 1817; 1822-1823; **1834**-1835; 1837; 1842; 1844; **1854**; 1881-**1882**; **1887**; **1889**-1890; 1896; **1907**; **1935**; **1943**; 1960; 1970-**1973**; **1975**; 1977; 1979 |



| | |
|---|---|
| Antigua<br>(period covered: 1770-1890) | 1770-**1771**; 1776-1780; 1782; 1788-1791; 1794; 1796-**1797**;<br>1804-1805; 1808; 1812; 1820-1822; 1829; 1833; **1834**-1837;<br>**1841**; **1845**; 1849-1850; **1856**; **1858**; **1860**; **1863**-1864; 1866;<br>1868; **1871**-1874; **1882**; **1885** |
| Guatemala City<br>(period covered: 1760-1945) | 1796; 1803; 1810; 1821-1822; 1824; 1829; 1840; 1843-1844;<br>**1860**; 1868; **1885**; **1888**; **1891**; **1895**; 1899; **1912**; 1914;<br>**1920**; **1922** |
| British Honduras (period<br>covered: 1765-1980) | 1771, 1797, 1826, 1828, 1834, 1848, 1852, 1854, 1856, 1858,<br>1859, 1860, 1863, 1865, 1867, 1871, 1872, 1882, 1884, 1885,<br>1886, 1887, 1889, 1891, 1892, 1893, 1894, 1895, 1901, 1903,<br>1907, 1908, 1909, 1912, 1916, 1917, 1919, 1920, 1921, 1922,<br>1923, 1924, 1926, 1927, 1928, 1932, 1933, 1935, 1937, 1938,<br>1939, 1940, 1941, 1944, 1945, 1946, 1947, 1948, 1949, 1950,<br>1951, 1952, 1953, 1955, 1957, 1958, 1959, 1962, 1963, 1964,<br>1967, 1973, 1974, 1975, 1976, 1980 |

The comparison reveals significant discrepancies between the four regions considered, with only a few periods consistently appearing as dry. The drought recorded in British Honduras in 1771 appears to have been part of an exceptionally severe and long event that affected the Mexican Yucatan Peninsula for ten years and impacted other Caribbean regions as well. During this period, Antigua experienced a drought in the early 1770s, while Jamaica faced extremely dry conditions between 1768 and 1773 (Chenoweth, 2003; Morgan et al., 2022). During the decade spanning

1800 to 1810, numerous droughts were documented especially in the Mexican Yucatan Peninsula and Antigua. However, there are no recorded references to such events in British Honduras. The severe drought of 1871 was reported in Antigua, but not in the Mexican Yucatan Peninsula, nor in Guatemala City. The records available confirm that the 1880s represented a very dry decade in the region. The severe drought in 1885 in British Honduras notably coincided with a similarly dry year in both Antigua and Guatemala City. Conversely, the dry events of the 1900s were not observed in the

other regions considered, except from the drought of 1907, which affected the Mexican Yucatan Peninsula as well as other Caribbean islands such as Jamaica, Cuba and Nassau (The Colonial Guardian, 1 June 1907; The Colonial Guardian, 15 June 1907). Similarly, the very dry conditions of the 1920s seem unique to British Honduras, with only Guatemala City experiencing two dry events early in the decade. The records from the Mexican Yucatan Peninsula, the only ones covering the late 20[th] century, do not indicate dry conditions between the mid-1940s and the mid-1950s as seen in British

Honduras, with the exception of the drought recorded in 1943. Finally, the drought that affected Belize in 1975 was also observed in the Mexican Yucatan Peninsula, which experienced severe and prolonged dry conditions in the first half of the 1970s.



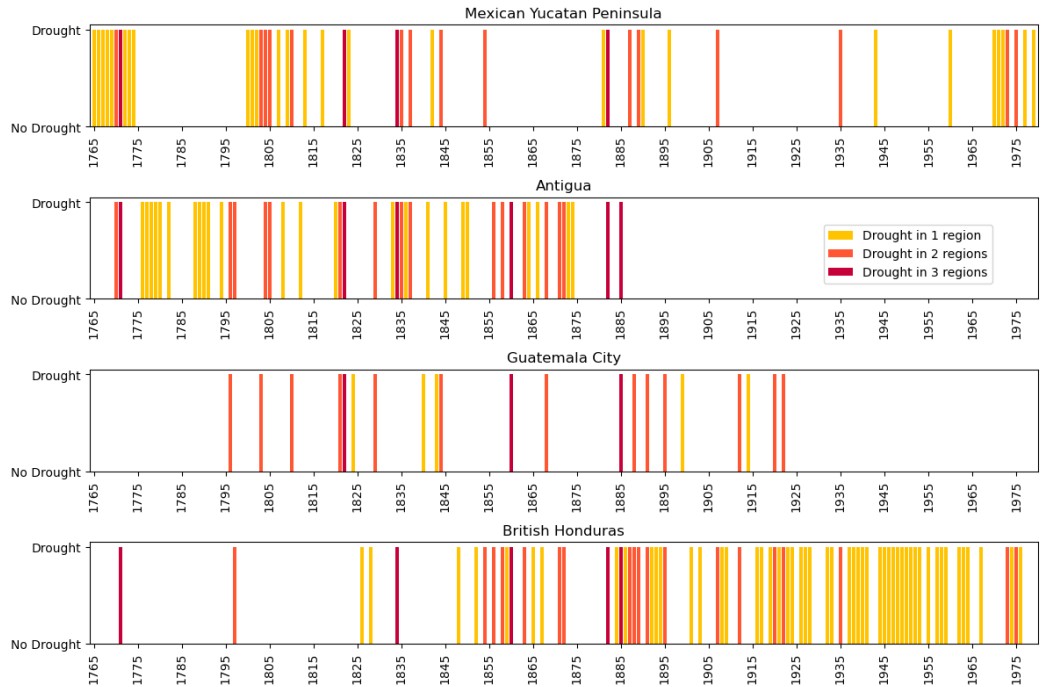

**Figure 8**: **Dry years in the Mexican Yucatan Peninsula, Antigua, Guatemala City and British Honduras from 1765 to 1980.** The yellow bars represent years when drought affected only one of these regions, while orange indicate years when two regions experienced dry conditions. Red bars indicate the years when droughts extended across three regions.

A final comparison of British Honduras droughts with those in the rest of Mexico reveals some expected discrepancies due to notable differences in response to key climate drivers (e.g. ENSO and AMO) between northern and central Mexico and the Yucatan Peninsula (Metcalfe et al., 2015; Stahle et al., 2016). Severe droughts and related famines affected both central and northern Mexico between the 1760s and 1785-1786. Dry conditions reappeared in the first decade of the 19[th] century, and again between 1811 and 1820 in the southeast. With the exception of the southeast, nearly all of Mexico experienced an exceptionally wet period from 1814 to 1817. This was followed by a severe drought from 1819 to 1823, which was especially intense in the northeast. Severe drought conditions affected central-northern Mexico from 1892 to 1896, with below-average rainfall persisting until 1915. The 1920s witnessed wetter conditions prevailing across the country. However, dry spells recurred from the 1940s through the early 1950s, notably intensifying from 1944 to 1946, mirroring conditions akin to those experienced in British Honduras. One of the most extreme periods of drought of the last 600 years is recorded in much of Mexico (focused in the north) over the period 1951-1957 (Stahle et al., 2016). Finally, after a decade of wetter conditions, the 1970s (particularly 1971-1975) experienced a general decline in rainfall with successive years of drought across various parts of Mexico, particularly in the southeast (O'Hara and Metcalfe, 1995; O'Hara and Metcalfe, 1997; Endfield, 2007; Stahle et al., 2016).

**7 Conclusions**





This article offers the first archival investigation of historical droughts in British colonial Belize. The documentary-derived chronology of droughts and the drought severity index obtained extend the existing climate record and provide an assessment of long-term drought patterns beginning in the late 18ᵗʰ century. This study demonstrates the value of historical records as a vital source for reconstructing past droughts. By incorporating confidence ratings and drawing from a diverse range of documentary sources—including official colonial reports, newspapers, logging activity records, and

travel accounts—this research has created a robust time series of drought events in Belize between 1771 and 1981 (independence). Where possible, discrepancies in historical records were addressed through the integration of precipitation data, enhancing the accuracy of the drought reconstruction. However, gaps remain in the drought chronology, primarily due to the limited availability of sources and the uneven occupation of the country. As a result, the chronology of droughts in northern British Honduras (Fig. 5) shows relatively few events recorded between the early

1770s and the mid-19ᵗʰ century, with a marked increase in dry anomalies from the late 1840s onwards, reflecting an increase in the availability of historical sources. Dry events in the south could only be traced from the early 1880s, as the region remained sparsely populated and produced few records until then (Fig. 6).

    The documentary-derived chronology of droughts has revealed that severe droughts affected parts of British Honduras during the years 1871, 1882-1889, 1895, 1901-1908, 1919-1927, 1943-1955 and 1975. The instrumental records generally

confirm the main dry events identified by historical sources, but several discrepancies are noted throughout the entire period considered. Notably, the precipitation series for Belize City does not corroborate the severe dry periods reported for the 1880s, 1901 and 1907. In contrast, the wet period recorded in the meteorological record from Stann Creek for 1943 to 1948 is not documented in the historical sources. The integration of this study's findings with drought reconstructions from the Mexican Yucatan Peninsula, Guatemala City, and Antigua reveals the occurrence of widespread

droughts across Central America and the Caribbean during the late 1760s to early 1770s, the 1880s, 1907 and the early 1970s. This comparison also highlights drought events that appeared to uniquely affect colonial Belize, such as the dry conditions of the 1920s and the period from the mid-1940s to mid-1950s, which were not documented in the Mexican Yucatan Peninsula, but in the case of the latter did occur in other parts of Mexico. This regional variation reflects the complexity of climatic drivers across this region (Metcalfe et al., 2015; Martinez et al., 2019; Holmes et al., 2023).

The study reveals both the strengths and limitations of using historical records and instrumental data for identifying droughts in British Honduras. Documentary records play an essential role in reconstructing past climates, particularly before reliable instrumental data became available. These sources often provide detailed descriptions of climate events, along with their short and long-term impacts, and can be highly precise in terms of both time and location. This precision is especially valuable in regions where meteorological stations were sparse or absent, as documentary evidence can

complement or fill gaps in instrumental data. However, this research has shown that documentary records tend to emphasise more severe events that affected large areas and had significant economic or infrastructural consequences. In contrast, shorter or more localised dry spells are often mentioned only briefly, if at all, because their impact was more limited in scope. The use of confidence ratings, as introduced by Kelso and Vogel (2007), further exacerbates this bias, as only climate events reported multiple times in historical sources are assigned a high confidence rating (CR 3). This

method inherently favours widespread, severe climate events that were well-documented due to their extensive effects. Conversely, brief or localised anomalies, which could obviously be mentioned by only a few sources—or in some cases, a single source—can only be assigned a lower confidence rating (CR 1).

    Precipitation series can help to mitigate this bias, as they typically provide monthly, and in some cases daily, rainfall data that can highlight even brief dry spells. However, meteorological stations in British colonial Belize were sparse and





widely distributed, particularly until the mid-20[th] century. In addition, the available precipitation records are often incomplete. Consequently, their spatial and temporal coverage is extremely limited, highlighting the necessity of integrating documentary sources to develop a comprehensive understanding of past climate anomalies.

Finally, this article's findings can be contextualised within both the longer term historical past and future climate projections through the analysis of mean annual precipitation reductions. Average precipitation reductions serve as

valuable proxies for comparing the severity of British colonial-era droughts in Belize with the multidecadal droughts that impacted the Maya Lowlands during the Terminal Classic Period (800-1000 CE). These droughts probably caused significant societal disruptions and contributed to the widespread collapse of the Classic Maya political systems (Kennett et al., 2012). Simultaneously, understanding the magnitude of past precipitation changes can inform assessments of projected future climate patterns in the region. An analysis of the available rainfall data from British Honduras indicates

that annual precipitation during the driest years on record decreased by approximately 32 % to 57 % in northern Belize and around 29 % to 59 % in the southern districts (Table 4). Data from the longest available time series were analysed to examine the average annual precipitation reduction during the driest recorded ten-year period at each station. The analysis reveals that mean annual precipitation decreased by up to 29.6 % in Corozal during 1918-1927 and reached a peak reduction of 32.6 % in Stann Creek between 1950 and 1959. In contrast, precipitation changes over a ten-year period were

more modest in the southernmost district of Toledo. The driest period, recorded at Punta Gorda between 1915 and 1924, showed a precipitation reduction of 10.7 % (Table 5).

**Table 4**: **Percentage decrease in precipitation during the driest years recorded across 13 meteorological stations in British Honduras.**

| Meteorological Station | Precipitation Reduction during Driest Year on Record (year) |
|---|---|
| Santa Rita Corozal | 35.94 % (1855) |
| Corozal | 56.54 % (1923) |
| Corozal Agricultural Station | 46.34 % (1949) |
| Orange Walk | 45 %    (1939) |
| Orange Walk Agricultural Station | 40.52 % (1959) |
| Yo Creek Agricultural Station | 32.68 % (1967) |
| Belize City | 47.2 %  (1923) |
| San Ignacio | 48.22 % (1923) |
| Central Farm Agricultural Station | 57.11 % (1949) |
| Stann Creek | 59.36 % (1959) |
| Stann Creek Agricultural Station | 42.58 % (1949) |
| Punta Gorda | 39.68 % (1944) |
| Punta Gorda Agricultural Station | 29.04 % (1975) |


Palaeoclimate records from the Yucatan Peninsula suggest that summer precipitation reductions during the Terminal Classic Period ranged from 30 % to 50 %, corresponding to annual mean precipitation decreases of 25 % to 40 %. Severely dry periods during this time may have seen precipitation reductions of up to 70 % (Medina-Elizalde et al., 2012; Evans et al., 2018). Speleothem records from Yok Balum in southern Belize support the occurrence of multidecadal droughts,

particularly between 820 and 870 CE, and reinforce the evidence for a ~40 % reduction in summer rainfall during the Terminal Classic Period (Kennett et al., 2012). Comparing precipitation data from the Terminal Classic and the British colonial period is challenging due to dating uncertainties and the limited climate resolution in paleoclimatic records.





However, it appears that while the driest years recorded in British colonial Belize show annual precipitation reductions not dissimilar to those experienced during the Terminal Classic, the duration of the droughts differs significantly. British colonial droughts rarely lasted more than a decade and never triggered the abandonment of human settlements. Furthermore, the average precipitation reductions over extended periods were notably lower than those observed in the 9[th] and 10[th] centuries.

Future climate projections for Belize indicate a decline in mean precipitation, particularly during the wet season. Notably, projections show substantial reductions in mean daily rainfall for July and an intensification of the midsummer drought (Imbach et el., 2018; Castellanos et al, 2022). Under the SSP3-7.0 scenario, annual average precipitation is expected to decrease by approximately 13 % by the end of the century, with reductions peaking at around 26 % during the June-July-August trimester (World Bank Group, 2024). Forecasts also indicate a 7-8 % shortening of the length of the wet season and a 6-8 % lengthening of the dry season by 2100 (Belize's Updated Nationally Determined Contribution, 2021).

Historical records and pre-modern instrumental records provide unique insights into the distribution, intensity and impacts of drought. Comparison of these droughts with both the deeper past, when drought was associated with major cultural disruption of the Classic Maya civilisation, and the possible future, bring new perspectives on how far future climate change may, or may not, be distinctly different from past experience in Belize.

**Table 5**: **Annual average precipitation reduction during the driest 10-year period recorded at 10 meteorological stations in British Honduras.**

| Meteorological Station | Annual Average Precipitation Reduction over a 10-year span (reference period) |
| --- | --- |
| Corozal | 29.63 %  (1918-1927) |
| Corozal Agricultural Station | 8.44 %   (1948-1957) |
| Orange Walk | 15.67 %  (1919-1928) |
| Belize City | 20.74 %  (1943-1952) |
| San Ignacio | 11.6 %   (1951-1960) |
| Central Farm Agricultural Station | 21.82 %  (1949-1958) |
| Stann Creek | 32.55 %  (1950-1959) |
| Stann Creek Agricultural Station | 11.45 %  (1944-1953) |
| Punta Gorda | 10.7 %   (1915-1924) |
| Punta Gorda Agricultural Station | 5.69 %   (1949-1958) |

**Data Availability**

Research data and codes used will be provided if paper is accepted for publication.

**Author contribution**

OAG and SEM developed the research idea. OAG carried out archival research and data collection. EACR contributed with data collection and article review. OAG and SEM lead the formal analysis. BDLBB contributed with maps preparation. GHE, FS, SM and AM contributed to the review and editing stages. OAG prepared the manuscript with contributions from SEM and all the other co-authors.

**Competing interests**



The authors declare that they have no conflict of interest.

**Acknowledgements**

The authors thank Mary Alpuche, Archives Officer I at the Belize Archives and Records Service in Belmopan, for her invaluable guidance in navigating the local collections and identifying relevant materials. Special thanks also go to the staff of the Munby Rare Books Room at Cambridge University Library for their assistance in locating historical records on 19th century British Honduras. Final thanks to Catherine Smith (Royal Holloway, University of London) for her advice on accessing the Belize archives and Forest Department sources.

**Financial support**

This research was funded by the Leverhulme Trust through RPG 2021-288.

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
