# Peer review of "Historical Droughts in British Colonial Belize (1771-1981)"

_EGUsphere, 2025_

## Author Response (AR1)

**Point-by-Point Reply to Referees' Comments**

**Referee 1:**

Thank you very much for your review and thoughtful comments. Below, we will address each of the points raised.

1: On Deforestation: You raise a good point, as British Honduras forests were subject to severe deforestation for at least 2 centuries. I did find several historical sources dated 1880s and late-19th century that related deforestation with increased drought frequency in the northern half of the country. However, such remarks were limited to this short period and sources were overall not enough to establish a clear relationship between deforestation and changes of the climate.

No relevant change made for lack of sources.

2: Table 1, "Reference period" refers to the chronological period used to calculate the average annual rainfall, and it is reported in brackets. I will add "(combined rainfall data)" to Belize City met station label in the table, thank you.

Relevant change made: added sentence as required to table

3: Figure 2, the dashed line is the precipitation average. I can change the caption describing this from "grey line" to "dashed line".

Relevant change made: "Grey line" changed in "dashed line" and caption describing dashed line introduced.

4: Figures 2-3, thank you for your comment, I will use the same scale of y axis in figure 3 (350 inches). I can also add in the caption of both figures that missing data points are due to lack of historical sources for specific years.

Relevant change made: Figure 3 has same scale on y axis (350 inches)

5: Figure 4, correct, this refers to each time different historical records mention a dry event in a year. On the threshold for drought: the article considers every abnormally dry event as drought, divided into 3 categories based on their severity and effects/damages. Figures 5 and 6 report every single dry event reported by sources and classifies them into the 3 categories. In sum, all of the events found are reported in the figures.

No relevant changes made on this comment

6: Figure 5, I completely understand your comment, the higher resolution of figure 5 is due to the higher number of events encountered for the northern region compared to those that affected the south. The long chronological period considered (1770-1990) and the numerous droughts of the 20th century inevitably make figure 5 a bit dense. I will add sub-subsections as suggested by Referee, maybe for every 20-30 or more years depending on the period (eg. 1770s-1790s; 1850s-1870s), just to avoid having too many sub-subsections.

Relevant change made: several sub-subsections were introduced as suggested by reviewer in section 5 of the article

Thank you very much,

Oriol Gali and co-authors

**Referee 2:**

Thank you very much for your review and thoughtful comments. Below, we will address each of the points raised.

1: All the instrumental data on colonial Belize known to the authors were used in the article. These were collected from both published and unpublished documents, including newspapers and colonial reports. The data used in the article include the few instrumental observations from Belize City airport reported by the Dominguez-Castro database (2017). These are actually incomplete if compared with the record we used in the article. Dominguez-Castro is using an early 20th century article and did not use archival materials. Brönniman et al uses the Dominguez-Castro article without adding any new instrumental observation.

Relevant change made: lines 116-118 (page 4) respond to the reviewer comment and refer to the Dominguez Castro article (not to Bronniman's as his article used Dominguez-Castro as model and did not introduce any new relevant data for our article). The new source was then, obviously, added to bibliography.

2: All precipitation data were given in both inches (the original unit of measurement used in British colonial Belize) and millimetres, with the exception of figures 2 and 3. Thank you for pointing this out, we will amend this.

Relevant change made: the text already reported all measurements in both units, with the exception of Figures 2 and 3. This was amended by adding a second y axis in mm.

3: Thank you for the comment, we will provide additional clarification in Figure 4 caption.

Relevant change made: both text highlighted by reviewer and caption of figure 4 were revised to improve clarity.

4: The use of categories of drought does carry some limitations, as every event had slightly different features, effects and impacts. However, the article uses categories of drought to help the reader identify patterns of drought in the long-term period. The categories used were created using the specific descriptions of the single events, so the categories reflect the conditions of British Honduras. Moreover, when instrumental data were available, droughts were also categorised according to precipitation levels reported by sources. By using these methods, the authors have tried to overcome the lack of objectivity mentioned.

Relevant change made: The above comment was added to section 4 on methodology (lines 280-287).

Thank you very much,

Oriol Gali and co-authors